# Anomalously rotary polarization discovered in homochiral organic ferroelectrics

Peng-Fei Li[1,*], Yuan-Yuan Tang[1,*], Zhong-Xia Wang[1,*], Heng-Yun Ye[1], Yu-Meng You[1] & Ren-Gen Xiong[1]

Molecular ferroelectrics are currently an active research topic in the field of ferroelectric materials. As complements or alternatives of conventional inorganic ferroelectrics, they have been designed to realize various novel properties, ranging from multiferroicity and semiconductive ferroelectricity to ferroelectric photovoltaics and ferroelectric luminescence. The stabilizing of ferroelectricity in various systems is owing to the flexible tailorability of the organic components. Here we describe the construction of optically active molecular ferroelectrics by introducing homochiral molecules as polar groups. We find that the ferroelectricity in (R)-(−)-3-hydroxlyquinuclidinium halides is due to the alignment of the homochiral molecules. We observe that both the specific optical rotation and rotary direction change upon paraelectric-ferroelectric phase transitions, due to the existence of two origins from the molecular chirality and spatial arrangement, whose contributions vary upon the transitions. The optical rotation switching effect may find applications in electro-optical elements.

[1] Ordered Matter Science Research Center, Southeast University, Nanjing 211189, China. * These authors contributed equally to this work. Correspondence and requests for materials should be addressed to H.-Y.Y. (email: hyye@seu.edu.cn) or to R.-G.X. (email: xiongrg@seu.edu.cn).

Optical rotation/optical activity is a phenomenon of rotation of plane-polarized light as it travels through chiral media, where the chiral media may be the chiral molecules or crystals. The rotatory direction may be either clockwise or anticlockwise, depending on the chirality of media. A racemic mixture containing equal amounts of right-handed and left-handed forms shows no optical activity. Such structure-property relationship makes optical rotation suitable for versatile applications, ranging from chiral analysis in chemistry, pharmaceutical and biological industries to electro-optical elements[1,2].

Ferroelectrics are crystalline materials with a variety of electrical and optical properties. As polar materials, ferroelectrics should crystallize in the ten polar point groups ($C_1$, $C_2$, $C_s$, $C_{2v}$, $C_4$, $C_{4v}$, $C_3$, $C_{3v}$, $C_6$, $C_{6v}$), seven of which, including five enantiomorphic groups ($C_1$, $C_2$, $C_4$, $C_3$, $C_6$,) and two non-enantiomorphic groups ($C_s$, $C_{2v}$), are optically active[3]. Because ferroelectric phase transitions and ferroelectric polarization reversal are accompanied by the change of the chirality of the crystals in many cases, ferroelectric optical activity is of importance for practical applications and the understanding of phase transition mechanisms[4–10].

Homochiral molecules form enantiomorphic crystals of the corresponding handedness. Compared with achiral compound, homochiral compounds are more likely to crystallize in the five enantiomorphic ferroelectric point groups. Therefore, it would be an efficient way to construct optically active ferroelectrics by introducing the homochiral molecules. Moreover, the introduction of molecular chirality can avoid the racemization of the bulk phases (because any noncentrosymmetric crystal crystallizes in the right-handed and left-handed forms with equal probability). The past few years have seen the rational design of molecular ferroelectrics to realize novel functions, and much progress has been achieved experimentally and theoretically[11–24]. However, the emerging molecular ferroelectric materials comprised of homochiral molecules are few and limited to those containing homochiral tartaric acids, such as cocrystals 1,4-diazabicyclo[2.2.2]octane $N,N'$-dioxide[25] and bis(imidazolium) L-tartrate[26]. No investigation has been performed on the optical activity changes during the ferroelectric phase transitions, although many other physical property changes near $T_c$ (Curie temperature, the phase transition temperature) are interesting during the ferroelectric phase transition. On the other hand, homochiral molecules are available only in organic compounds, whereas none is found in inorganic oxides including inorganic ceramic ferroelectrics. Consequently, there is much room to explore molecular ferroelectrics containing homochiral organic molecules, where this idea can be back to the first ferroelectric Rochelle salt discovery (a homochiral organic salt)[27].

In this context, we designed above-room-temperature molecular ferroelectrics containing homochiral molecules: $(R)$-$(-)$-3-hydroxlyquinuclidinium halides (Fig. 1). Although similar molecules like quinuclidine and 1,4-diazabicyclo[2.2.2] octane have been found to be excellent units for construction of ferroelectrics owing to their spherical geometries, which easily lead to structural phase transitions and allow the reorientation of the molecules[16,25,28,29], the chirality of these molecules has never been noticed to be useful for generation of ferroelectricity and optical activity. We observe that the optical activity of $(R)$-$(-)$-3-hydroxlyquinuclidinium halides near $T_c$ shows significant changes in both specific optical rotation and the rotatory direction upon the ferroelectric phase transitions. This phenomenon is unprecedented as far as we know. We here describe their optical rotation, ferroelectric and related properties.

## Results

**Structural phase transition.** The structural phase transition is one of the most important properties for understanding the ferroelectricity; we first determined the crystal structures of the ferroelectric and paraelectric phases (abbreviated as FP and PP respectively). $(R)$-$(-)$-3-hydroxlyquinuclidinium halides are iso-structural, and their optical and ferroelectric properties are similar. Therefore, we here take $(R)$-$(-)$-3-hydroxlyquinuclidinium chloride (**1**) as an example to describe their structures and properties, whereas the related information for $(R)$-$(-)$-3-hydroxlyquinuclidinium bromide (**2**) and iodide (**3**) is

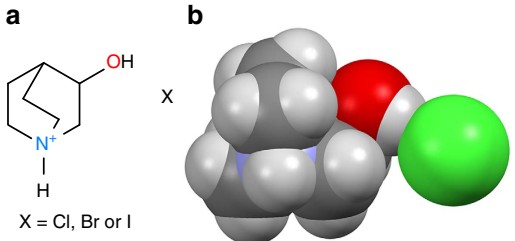

**Figure 1 | Chemical structures of $(R)$-$(-)$-3-hydroxlyquinuclidinium halides.** (**a**) Structural formula of $(R)$-$(-)$-3-hydroxlyquinuclidinium halides. (**b**) The space-filling drawing of $(R)$-$(-)$-3-hydroxlyquinuclidinium chloride (**1**), showing the globular geometries of both the cation and the anion.

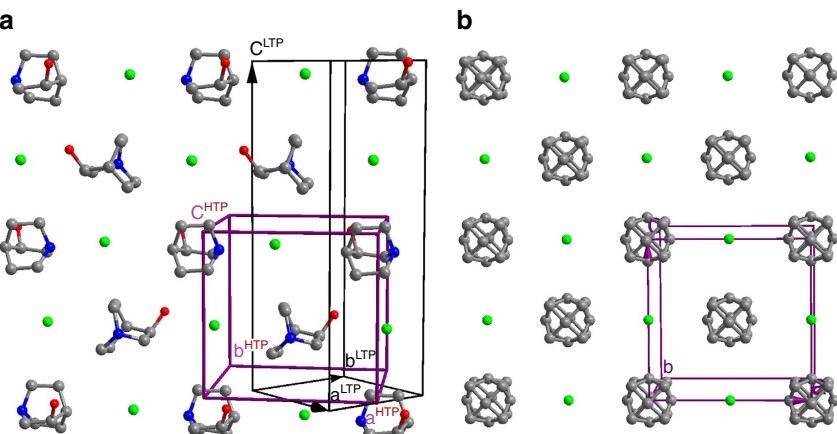

**Figure 2 | The comparison of the crystal structures of the FP and PP.** (**a**) A view of a [1 1 0] layer of the FP. The cell edges of both the FP and PP were drawn to show their relationship. (**b**) A view of a [0 1 0] layer of the PP.

supplemented in the Supplementary Material. At room temperature, **1** crystallizes in the chiral polar space group $P4_1$, $a = 6.7217(10)$ Å, $c = 18.464(4)$ Å, as reported in the literature[30]. The crystal consists of the homochiral $(R)$-$(-)$-3-hydroxlyquinuclidinium cations and $Cl^-$ ions, which are involved in the head-to-tail one-dimensional hydrogen bonded chains through the $O–H^{acid}\cdots Cl$ and $N–H^{acid}\cdots Cl$ interactions along the $a$- or $b$-direction (Supplementary Fig. 1). The crystal packing structure looks like that of the familiar FCC (face-centred cubic) NaCl, that is, the cations and anions are arranged alternatively along the [1 1 0], [1 −1 0] and [0 0 1]-

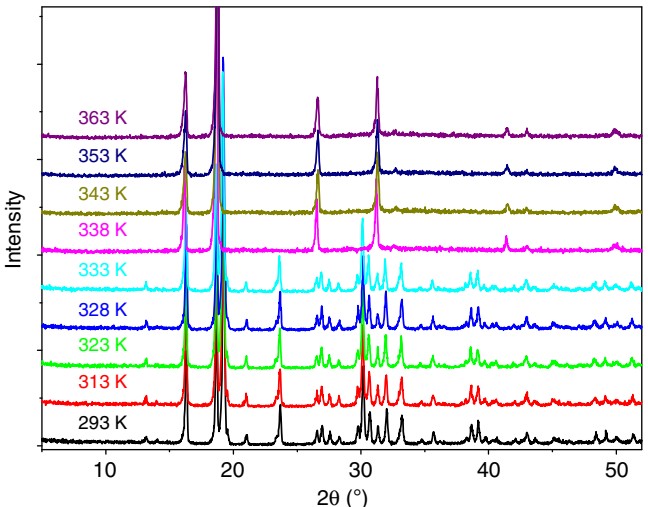

**Figure 3 | Temperature-dependent PXRD of 1.** The patterns of PXRD were recorded in a heating run. The significant change of the pattern was observed at 338 K.

directions (Fig. 2a). The determined absolute configuration of the hydroxlyquinuclidinium cation is consistent with that of the starting material.

The thermal analysis on the polycrystalline sample of **1** reveals good thermal stability up to 485 K (Supplementary Fig. 2) and a high temperature structural phase transition at around $T_c = 340$ K (Fig. 4a). As will be described below, the phase below $T_c$ is the FP, the phase above $T_c$ is the PP. To probe the structural information of the high-temperature PP, both the variable-temperature single crystal and powder X-ray diffractions were measured. Patterns of the PXRD (powder X-ray diffractions) below $T_c$ match well with the simulated one from the single crystal structure, revealing the high crystallinity and purity of the phase (Supplementary Fig. 3). Many diffractions peaks observed at below $T_c$ disappear upon heating above $T_c$ (Fig. 3). The smaller number of peaks observed means that the symmetry of the PP becomes very high. The PXRD data at 363 K were refined with the Pawley method (Supplementary Fig. 4). An FCC unit cell with $a = 9.4666$ Å was suggested, and the most possible space group is the $F432$.

The high temperature single crystal X-ray diffraction shows few peaks with relatively weak intensity especially for those in the relatively high-angle region. We notice that this situation is very similar to that recently observed for the quinuclidinium salt which undergoes a ferroelectric transition[28]. In this case, the high temperature phase becomes a cubic plastic phase. At the transition point, the isotropic cubic phase changes to a polar rhombohedral phase. In our case, the[31,32] high-temperature structure was refined in the space group $F432$, $a = 9.5084(18)$ Å. The pattern of the PXRD matches well with the simulated one from the single crystal structure, supporting the structure model (Supplementary Fig. 4). The relationship of the two temperature cells is $\mathbf{a}^{PP} \approx \mathbf{a}^{FP} + \mathbf{b}^{FP}$, $\mathbf{b}^{PP} \approx -\mathbf{a}^{FP} + \mathbf{b}^{FP}$, $\mathbf{c}^{PP} \approx 0.5\mathbf{c}^{FP}$ (Fig. 2a). The $4_2$ fourfold screw axis of the PP along the $c$-direction becomes the $4_1$ fourfold screw axis of the FP, because of the doubling of the $c$-axis upon the transition from the PP to FP, and

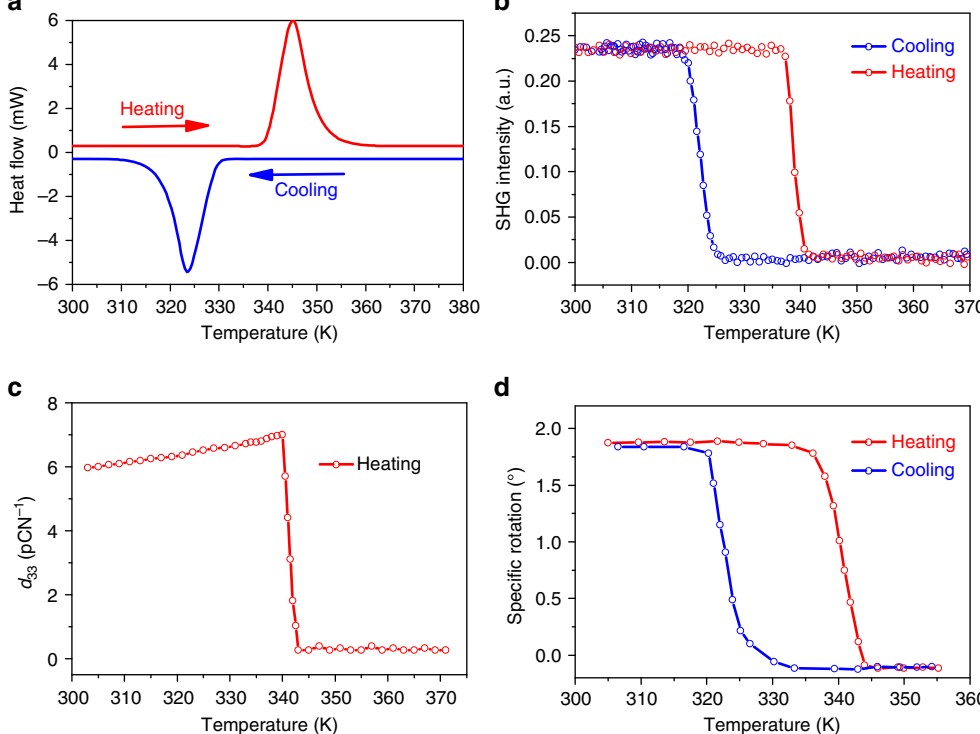

**Figure 4 | Phase transition properties of 1.** (**a**) The DSC (differential scan calorimetry) curve. (**b**) The temperature-dependence of the SHG signal. (**c**) The temperature-dependence of the piezoelectric coefficient $d_{33}$. (**d**) Temperature-dependent specific rotation of a crystal through the optical axis.

accordingly, the origin of the FP cell is at the positon (0.25, 0.25 0) of the PP cell. The crystal packing of the PP is similar to that of the FP (Fig. 2b). The cation is located at the crystallographic special position 432, and thus is severely disordered and molded with a spherical structure. It can be considered to undergo a high degree of dynamic reorientation (rotation). This suggests that the PP is a plastic phase in which the molecules are nearly freely rotating[31,32]. We recorded the conductivity of **1** in the temperature range from 300 to 373 K (Supplementary Fig. 5). The conductivity indeed changes from room-temperature $4.27 \times 10^{-6}\,\mathrm{S\,m^{-1}}$ to high-temperature $1.99 \times 10^{-4}\,\mathrm{S\,m^{-1}}$ (at the level for those of molecular-ionic plastic conductors[33]) at around $T_c$, verifying the nature of the plastic phase transition.

**Second harmonic generation effect**. The SHG (second harmonic generation) property is one of the physical properties which depend on the crystalline symmetries of crystals[3]. As an even-order nonlinear optical effect, it is only allowed in crystals without inversion symmetry. The measurement of SHG has been a powerful method of testing crystalline materials for the absence of the symmetry centre[34–36]. As shown in Fig. 4b, the SHG signal of **1** undergoes a clear transition from non-zero to zero intensity at around $T_c$. The SHG-active and SHG-inactive states are consistent with the symmetry requirements of the point group 4 of the FP and the point group 432 of the PP, respectively.

**Piezoelectric effect**. The piezoelectric effect is the similar physical property which depends on the crystal symmetry. We thus also measured piezoelectric effect in the temperature range from room temperature to 373 K. As expected, the $d_{33}$ undergoes a transition from the non-zero to zero at around $T_c$, consistent with the symmetry requirements of the PP and FP (Fig. 4c).

**Optical activity**. Both the solid state and the solution of **1** should be optically active because of the existence of homochiral $(R)$-$(-)$-3-hydroxlyquinuclidinium cations. We here focus on the optical activity in the solid state. As shown in Fig. 4d, strong optical rotation activity was observed for the single crystal at room temperature due to the helical arrangement of the $(R)$-$(-)$-3-hydroxlyquinuclidinium cations (Supplementary Fig. 1). The rotation direction is dextrorotatory (or clockwise rotation) when the linearly polarized light (589.3 nm) passes through the [0 0 1] direction of the bulk single crystal. The room-temperature optical rotation of **1** is about 1.88° for a 600 μm thick transparent single crystal, that is, $\rho = \varphi/d = 3.13\,^{\circ}\mathrm{mm^{-1}}$. According to the Fresnel's explanation, the small difference in the refractive indices $(\Delta n = n_l - n_r)$ between left-handed and right-handed circular polarizations quantifies the strength of the optical activity. For a dextrorotatory single crystal, $\Delta n$ is positive, $n_r < n_l$. The phase difference of the two circular polarized light can be expressed as $\Delta \Phi = 2\pi/\lambda_0(n_l - n_r)d$, where $\lambda_0$ is the wavelength of incident light in vacuum. $\Delta n$ of **1** can be estimated as follows: $\Delta n = \rho \bullet \lambda_0/\pi = 5.9 \times 10^{-4}$, which falls in the normal range for common optically active crystals, and is much smaller than the birefringence $(\Delta n = n_e - n_o)$ of birefringent crystals (usually $|n_e - n_o| = 10^{-3} \sim 10^{-1}$). The optical activity keeps constant upon heating until an abrupt change at around $T_c$. In the PP, it become levorotatory $(\Delta n < 0)$ and relatively weak. In the subsequent cooling process, the optical activity returns to the dextrorotatory state with a 20 K hysteresis. Interestingly, the optical axis sometimes changes upon cooling from the PP to FP. This is because the $c$-direction of the FP can be formed along any of the $a$-, $b$ and $c$-directions of the PP upon the transition from the space group $F432$ to $4_1$.

In general, crystalline optical activity arises from a combination of the chirality of the molecules and that of their non-centrosymmetric spatial arrangement[37]. In the FP, the inconsistence of the rotatory direction of the crystal with that of the molecule indicates that the contribution from the helical packing in the point group 4 is opposite to that from the molecule, and the former is greater. Probably, the polarization of the crystal (see below) plays an important role in the optical rotation, as observed in other ferroelectrics[4–10]. In the PP, the observed weak left-handed rotation indicates that the optical activity is mainly contributed by the molecules, while the contribution from the spatial arrangement is negligible. Such a state requires the high disorder of the molecular orientation. This is consistent with the assignment of the PP as the plastic phase.

**$P$–$E$ loops**. **1** is a simple organic salt. The center of the positive charge of the cation is located on the protonated N atom, and the negative charge is located on the $Cl^-$ ion. As shown in Supplementary Fig. 1, there is an obvious displacement between the positive and negative charge along the $c$-axis. According to the point charge model, the spontaneous polarization of **1** at room temperature was estimated to be 2.4 μC cm$^{-2}$, which is comparable to the typical molecular ferroelectric TGS (2.8 μC cm$^{-2}$) and much bigger than that of Rochelle salt (0.25 μC cm$^{-2}$). Polarization reversal was firstly investigated on a thin film capacitor with

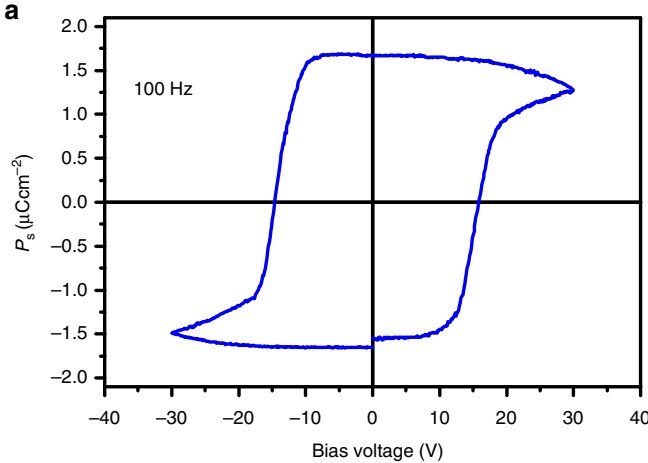

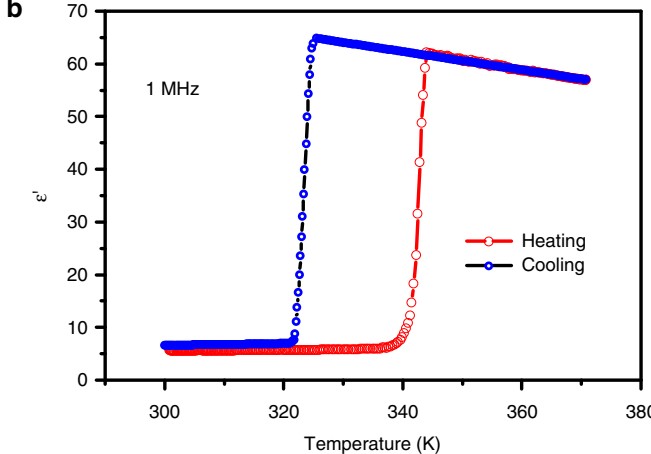

**Figure 5 | Polarization reversal and dielectric properties in 1.** (**a**) The electric polarization versus electric field ($P$–$E$) hysteresis loop measured from a film of 150 nm in thickness. (**b**) The real part $\varepsilon'$ of the complex dielectric constant ($\varepsilon$) ($\varepsilon = \varepsilon' - i\varepsilon''$), where $\varepsilon'$ is the real part, and $\varepsilon''$ is the imaginary part) as a function of temperature.

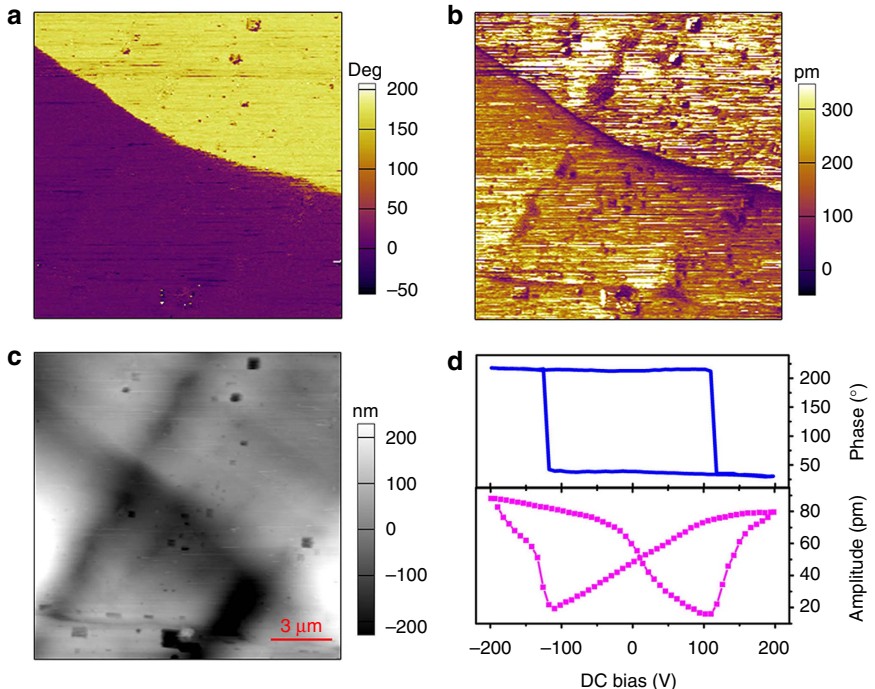

**Figure 6 | Domain structures observed for an as-grow crystal of 1.** (**a**) The vertical PFM phase image. (**b**) The vertical PFM amplitude image. (**c**) The topographic image of the crystal surface. (**d**) Phase and amplitude signals as functions of the tip voltage for a selected point, showing local PFM hysteresis loops.

the configuration of Au/Sample film/ITO (indium tin oxide glass). As shown in Fig. 5a, a typical electric polarization versus electric field (*P–E*) hysteresis loop was recorded at room temperature under a triangular AC electric field at 100 Hz. For a thin film of 150 nm in thickness, the coercive voltage for the polarization reversal is about 15 V, corresponding to a coercive field of 1 MV cm$^{-1}$. Such a high coercive field is due to that the polarization reversal involves a great amplitude of reorientation of the cations. The measured remnant polarization $P_r$ at 100 Hz is about 1.7 μC cm$^{-2}$. This is smaller than that calculated from the point charge model, due to the polycrystalline nature of the thin film.

**Dielectric properties**. We measured the complex dielectric constant as a function of the temperature in the frequency range of 5 to 1,000 kHz. The dielectric response at the frequency of 1 MHz is presented in Fig. 5b. As expected, the dielectric constant undergoes a large jump at around $T_c = 340$ K, and the dielectric constant at 342 K at each selected frequency is about ten times as large as that at 337 K, pointing to a ferroelectric transition. In general, the dielectric anomaly is a $\lambda$-shape peak for a second-order ferroelectric phase transition, while it is a step-like jump for a first-order transition. For **1**, the step-like jump of the dielectric constant reveals the character of the first-order transition, consistent with that in the differential scan calorimetry (DSC) results.

**Domain structures and polarization reversal**. The polarization reversal was further investigated by piezoresponse force microscopy (PFM). It can offer non-destructive visualization and operation of ferroelectric domains at the nanometer scale[38–42]. A PFM image contains two components: (i) a vertical PFM image, examined by recording the tip deflection signal at the frequency of modulation; and (ii) a lateral PFM image, measured as lateral motion of the cantilever because of bias-induced surface shearing. Each component can be characterized by the two parameters: amplitude and phase. The amplitude is proportional to the

reciprocal $d_{33}$, providing the information on the magnitude of polarization, and the phase presents the direction of the polarization. The measured samples include both the film and bulk crystal of **1**, and the results are consistent. The results on static domain structures for the bulk crystal samples are shown in Fig. 6. The PFM images were recorded along the polar *c*-axis. Therefore, the signal in the vertical PFM mode are strong, while the signal in lateral PFM response is negligible. The phase image exhibits two antiparallel domains with a clear contrast (Fig. 6a). As distinctly presented in the amplitude image (Fig. 6b), the domains are separated by the domain wall, which is irrelevant to the topography. The domains are so big that the shape of the domains could not be recorded.

The results on domain structures for the thin film sample are shown in Fig. 7. The thin film is the aggregates of thin single-crystal grains. The domains of thin film are smaller, and thus the shape can be observed. As shown in Fig. 7a,b,d,e, some domains have the almost tetragonal shape, in good agreement with the growth habit of the tetragonal crystal. The typical vertical and lateral PFM images reveals the same domain distribution in the two PFM mode. The dependence of phase and amplitude on $V_{DC}$ (Figs 6d and 7f), measured at a selected point, display a hysteresis loop and a butterfly curve respectively, which are typical evidence for the switching of ferroelectric domains[38–42]. The local coercive voltage of the crystal is much greater than that of film, as indicated by the minima of the amplitude loop.

The polarization reversal is reproducible, and the process at a selected region on the surface of the film is recorded in Fig. 8. Firstly, we scan the vertical PFM signals of initial state over an area of $15 \times 15$ μm$^2$, where the piezoresponse is basically homogeneous in both phase and amplitude images, suggesting a single-domain state of this area (Fig. 8a–c). Then the selected region was polarized with an electrically biased PFM tip: the film surface was scanned with a tip bias of $-60$ V exceeding the coercive voltage (Fig. 8d–f). The 180° phase contrast indicates that the polarization is antiparallel in the two domains. The

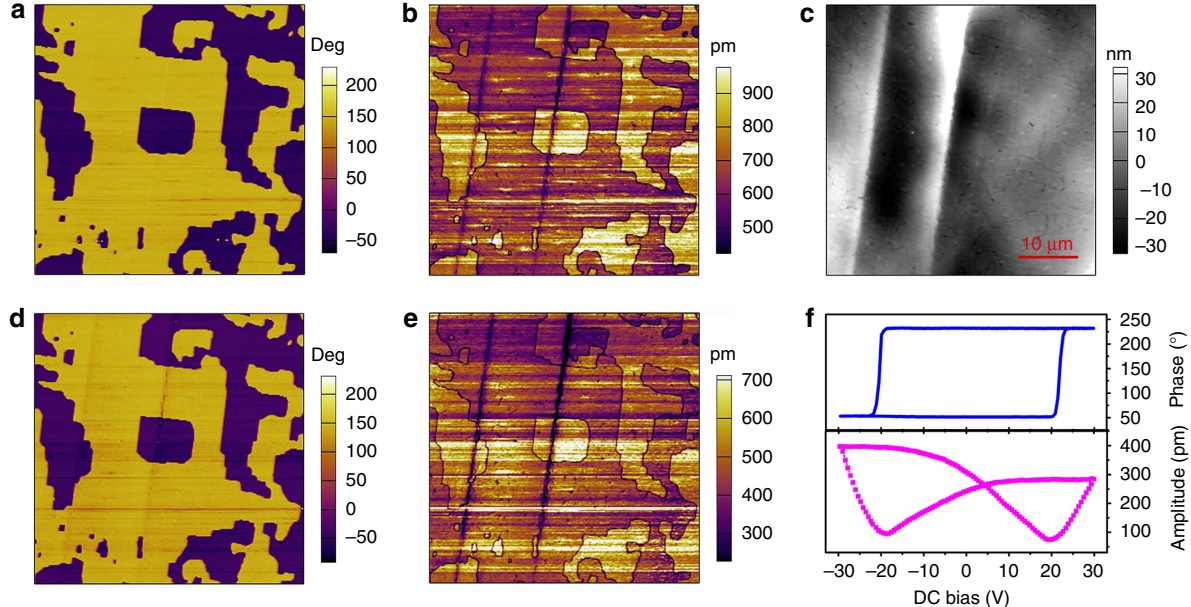

**Figure 7 | Domain structures observed for a thin film of 1.** (**a**) The vertical PFM phase image. (**b**) The vertical PFM amplitude image. (**c**) The topographic image of the film surface. (**d**) The lateral PFM phase image. (**e**) The lateral PFM amplitude image. (**f**) The phase and amplitude signals as functions of the tip voltage for a selected point, showing local PFM hysteresis loops.

**Figure 8 | Polarization reversal of 1 measured from a film.** The panels in each column are arranged as the sequence: the topographic image (up), the vertical PFM amplitude image (middle) and the phase image (bottom) of the surface. (**a–c**) Images for the initial state. (**d–f**) Images for the state after the first switching operation, produced by scanning with the tip bias of − 60 V. (**g–i**) Images for the state after the succeeding back-switching operation, produced by scanning with the tip bias of + 60 V. (**j–l**) Images for the state after 18 h. The purple and yellow regions in phase images indicate the regions with polarization oriented upward and downward respectively.

purple and yellow regions in the phase image represent the regions with polarization oriented upward and downward, respectively. After a back-switching field with opposite bias of $+60$ V was applied to the tip, the polarization direction of a selected region in the centre can be switched back (Fig. 8g–i). It should be noted that the topographic image of the same area of the film, obtained after these processes, shows no sign of surface deformation. This observation confirms that the domains in the film surface of **1** can be switched by an external field. As depicted in Fig. 8j–l, the multi-domain structure remains unchanged when it was examined after 18 h later. The PFM results reveal the switchable and stable polarization in **1**, which is the characteristic feature of a ferroelectric material, distinguishing them from pyroelectric and piezoelectric materials.

## Discussion

In general, it is not easy to determine the space group of a plastic phase reliably because of the above-mentioned X-ray diffraction characteristics. In our case, the space group determined from the X-ray diffraction is just the one which satisfies symmetry requirement for the ferroelectric phase transition. The FP has the point group 4. According to Aizu rule (see Table I in ref. 43), there are only two possible paraelectric point groups: 422 and 432. Since both the powder and single crystal X-ray diffractions clearly reveal an FCC lattice of the PP, the paraelectric point groups should be 432. In addition, the observation that the optical axis sometimes changes upon the transitions of FP→PP→FP excludes the point group 422 as the paraelectric point group, because the 422F4 ferroelectric species is uniaxial, and the polar axis, which is just the optical axes, cannot change upon the transitions. Contrary to the 422F4 ferroelectric species, the 432F4 ferroelectric species is multiaxial, which allow the change of the polar axis (optical axis) upon the phase transitions. There are just two space groups which belong to the point group 432 with the FCC lattice: $F432$ and $F4_132$. It is worth noting that the $c$-axis of the PP is the doubling of that of the FP. This requires that PP should have the $4_2$ fourfold screw axis, not the $4_1$ fourfold screw axis. Only the space group $F432$ contains the $4_2$ fourfold screw axes. The $4_2$ fourfold screw axis of the space group $F432$ becomes the $4_1$ fourfold screw axis of the space group $4_1$ upon the transition from the PP to FP, which determines not only the $\mathbf{c}^{FP}$, but also the $\mathbf{a}^{FP}$ and $\mathbf{b}^{FP}$ as well as the origin which is located on the $4_1$ fourfold screw axis. That is why the relationship of the two temperature cells is $\mathbf{a}^{PP} \approx \mathbf{a}^{FP} + \mathbf{b}^{FP}$, $\mathbf{b}^{PP} \approx -\mathbf{a}^{FP} + \mathbf{b}^{FP}$, $\mathbf{c}^{PP} \approx 0.5\mathbf{c}^{FP}$ (Fig. 2a). Therefore, the crystal of **1** is of the 432F4 ferroelectric species among the 88 ferroelectric species[43].

Structurally, to realize the polarization reversal, the homochiral (R)-hydroxlyquinuclidinium cations should reorient upon applying a field since all the cations align along the $c$-axis (Fig. 2a and Supplementary Fig. 1). The reorientation involves relatively great amplitude of the motion of the cation. As is known, three-dimensional (3D) spherical molecular structures like tetramethylammonium, damantane 1,4-diazabicyclo[2.2.2] octane tend to exhibit dynamical disorder in the close packed crystals because of weak van der Waals interactions in the crystal lattice. Upon an H atom is replaced by an amino or hydroxyl group, the spherical symmetry is modified just a little (Fig. 1b). Accordingly, the molecules can undergo the plastic transition upon heating. Such a feature indicates that they can reorient upon an external field at room temperature. The homochiral (R)-hydroxlyquinuclidinium is the case. At the high temperatures, homochiral (R)-(−)-hydroxlyquinuclidinium are nearly freely rotating, which allows high symmetry of the PP and results in the cancelling of the molecular dipoles in any direction. At lower temperatures, homochiral (R)-hydroxlyquinuclidinium becomes ordered, the alignment of the dipole moments leads to

the spontaneous polarization. Therefore, the ferroelectric origin is due to the alignment of the chiral cations. This ferroelectric mechanism is different from that of the known ferroelectrics containing homochiral molecules, including Rochelle salt[44], the cocrystal of 1,4-diazabicyclo[2.2.2]octane $N,N'$-dioxide L-tartrate[25] and bis(imidazolium) L-tartrate[26]. In those ferroelectrics, the polarization reversal does not involve the reorientation of the homochiral L-tartrate acid because it has the intramolecular (pseudo) two-fold rotation axis. In their PPs, the dipoles are related by the two-fold axis; in their FP, the loss of the two-fold rotation symmetry leads to the spontaneous polarization.

In order to investigate the microscopic ferroelectric polarization, we carried out density functional calculations based on the Berry phase method developed by Kingsmith and Vanderbilt[45,46]. The continuous evolution of spontaneous polarization from the nonpolar structure ($\lambda=0$) to the polar structure ($\lambda=1$) is plotted as a function of dimensionless parameter $\lambda$ in Fig. 9a. From the curve, the estimated value is $\sim 4.94\,\mu C\,cm^{-2}$ in the ferroelectric structure, which are contributed mainly by the relative displacement between the positive and negative charge centres. This value is greater than that calculated from the point charge model. This may be due to the additional considerations of molecular dipoles and electron polarization in the density functional calculation. The calculated polarization vector coincides with the crystallographic $c$-axis, which is in good agreement with the polar axis of the space group $P4_1$. In addition, the total energy gradually increases through the path towards the centric state (Fig. 9b), indicating that a compensation for the potential barrier with 0.72 eV is theoretically needed to reverse the spontaneous polarization in each unit cell.

In summary, we design molecular ferroelectrics by introducing the homochiral molecule (for more information on the structures and physical properties of **2** and **3**, see Supplementary Figs 6–10). The ferroelectricity is realized by the alignment of the homochiral molecule, owing to the spherical structure. This method can be used to construct more new molecule-based ferroelectrics, since there are a few known homochiral molecule with the spherical

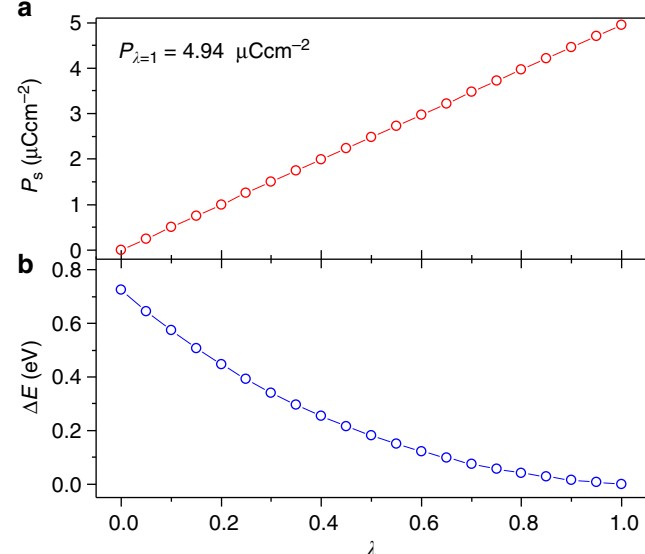

**Figure 9 | Results from the first-principles density-functional theory of 1.** (**a**) The variation of ferroelectric polarization along the path from the centric ($\lambda=0$) to the ferroelectric structure ($\lambda=1$). (**b**) The variation of the total energy in each unit cell along the path from the centric ($\lambda=0$) to the ferroelectric structure ($\lambda=1$). The dimensionless parameter $\lambda$ is the normalized amplitude of the atomic distortion connecting the structure of the PP and the structure of the FP.

structures. All investigated physical properties are consistent with the structural assignment. Among them, the change of optical axis in the phase transitions is due to the multiaxial ferroelectricity for the 432F4 ferroelectric species. The observed optical rotation switching effect may find applications in electro-optical elements. This finding would throw light on the rational design of homochiral molecular ferroelectrics for special purposes such as molecular recognition and chiral induction, some of which are not available from conventional inorganic oxide ferroelectrics.

## Methods

**Synthesis.** Plate-like colourless crystals of **1** were easily obtained by reaction of *R*-3-quinuclidinol with 36.5% aqueous HCl and recrystallized in acetonitrile-methanol mixed solvent. **2** and **3** are prepared using the same procedure. The purity of the bulk phase was verified by PXRD (Supplementary Fig. 3). For crystallographic information, see Supplementary Data 1–4, Supplementary Table 1 and Supplementary Figs 11–14.

**Optical rotation measurements.** Temperature-dependent specific rotation was measured with a polarimeter (Model: WZZ-2S) equipped with a home-made heating module. To be specific, a single crystal of $<001>$ slice ($\sim 600 \mu m$ in thickness) was mounted in the light path with polarized light parallel passing through the optical axis of the sample. With this configuration, influence of crystal birefringence can be eliminated. A sodium lamp with the wavelength of 589.3 nm was used as the light source.

**P–E measurements.** A thin-film crystal capacitor was fabricated for the ferroelectric hysteresis loop measurement. The detailed process is as follows. Firstly, commercial ITO-coated glass substrate ($\Phi = 15$ mm) was ultrasonically cleaned in toluene, acetone, ethanol and deionized water sequentially for 20 min at a time. A typical spin-coating process (6,000 RPM, 60 s) was employed with the precursor methanol solution of **1** (185 mg ml$^{-1}$). During the spin-coating, the large area single crystals were *in situ* grown on the ITO-coated glass with a perfect uniform coverage rate. A low temperature annealing process (35 °C, 30 min) was carried out to remove residual solvents. The film thickness was measured using a manmade gap using atomic-force microscopy (AFM). The top electrodes were then fabricated through sputtering gold target with a metal shadow mask ($\Phi = 1$ mm). Gold sputtering was conducted within ten cycles with a period of time of 30 s to minimize the thermal damage of the film. The *P–E* measurements were performed on a probe station equipped with a Precision Premier II Ferroelectric Tester (Radiant Technologies).

**PFM measurements.** Single crystals and thin films were used for the PFM measurements. The thin film was *in-situ* grown on ITO-coated glass as follows. For the 1 μm thick film, the solution of **1** (20 μl 185 mg ml$^{-1}$) was dropped on an ITO-coated glass substrate, and then allowed to crystallize via the slow evaporation of the methanol. For the 150 nm thick film, typical spin-coating method was utilized with 6 K RPM for 60 s. PFM visualization of the ferroelectric domain structures were carried out using a commercial atomic force microscope system (MFP-3D, Asylum Research). Conductive Pt/Ir-coated silicon probes (EFM-50, Nanoworld) were used for domain imaging and polarization switching studies. Resonant-enhanced PFM mode was used to enhance the signal, with a typical ac voltage frequency of about 360 kHz and ac amplitude of 1.0 V.

**Physical properties measurement.** Methods of DSC, SHG and dielectric measurements were described elsewhere[47,48]. For dielectric measurements, single-crystal plates with about 20 mm$^2$ in area and 0.5 mm in thickness were cut from the large crystals in the [0 0 1] direction. Silver conduction paste deposited on the plate surfaces was used as the electrodes.

**Theoretical calculation.** The first-principles calculations were performed within the framework of density functional theory implemented in the Vienna ab initio Simulation Package (VASP)[49,50]. The energy cut-off for the expansion of the wave functions was fixed to 550 eV and the exchange − correlation interactions were treated within the generalized gradient approximation of the Perdew − Burke − Ernzerhof type[51]. For the integrations over the k-space, we use a $4 \times 4 \times 1$ k-point mesh. In order to evaluate the ferroelectric polarization, we consider the path connecting the non-polar to the polar structure by linearly interpolating the atomic positions. The experimental room temperature crystal structure was used as the ground state for evaluating the ferroelectric polarization.

**Data availability.** The crystal structures have been deposited at the Cambridge Crystallographic Data Centre (deposition numbers: CCDC 1497196–1497198 and 1509639), and can be obtained free of charge from the CCDC via www.ccdc.cam.ac.uk/getstructures.

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

## Acknowledgements

This work was supported by 973 project (2014CB932103) and the National Natural Science Foundation of China (21290172, 91422301, 21427801 and 21573041).

## Author contributions

Z.-X.W., Y.-Y.T. and P.-F.L. prepared the samples and characterized the properties. H.-Y.Y. determined the structures. Y.-M.Y. and R.-G.X. wrote the manuscript, respectively. R.-G.X. designed and directed the studies.

## Additional information

**Publisher's note**: 

