## [Peer Review File · Nature Communications]

Reviewers' comments:

Reviewer #1 (Remarks to the Author):

This manuscript describes the optical rotation switching in homochiral organic ferroelectrics. The main point of this manuscript may be the finding of the optical activity change at the ferroelectric transition. The authors presented the structural, optical, and dielectric data, but I think that there are several serious faults and insufficiency in their interpretations.

1. The powder X-ray diffraction of HTP shows sharp peaks even in the relatively high-angle region. Smaller number of peaks observed means that the symmetry of the unit cell becomes very high. What I could not really understand is why the authors did not attempt to index the powder pattern. For such a simple pattern, it is rather easy to index the peaks. I think that the authors should index the pattern and determine the lattice parameters (crystal system, cell volume and the Z value) based on the powder diffraction of HTP. This information is extremely important for further discussion.

2. If HTP has a high-symmetry lattice and contains "severe disorder", the molecules are considered to undergo a high degree of dynamic reorientation (rotation). This suggests that the HTP is a plastic phase in which the molecules are nearly freely rotating. For such a case, electron density becomes featureless (spherically distributed electrons at the lattice sites), and the molecular framework and chirality may be vanished. Enantiomorphic point groups may be applicable only when the chirality in the lattice is warranted (This is the case when the site symmetry derived from the point 1 is consistent with the molecular chirality. In this case, electron density can be reproduced by superimposing multiple orientations of the molecular framework (restricted reorientation)). Because the authors should discuss the structure of HTP taking the results obtained from the point 1 into account, the present point group assignment for HTP being persistent in non-centrosymmetric may be fundamentally wrong.

3. Though the authors mention "the SHG signal undergoes a clear transition from non-zero to zero intensity" in p. 7, the SHG intensity in HTP shown in Figure 2b is not zero. This made the reviewer in confusion. If the SHG intensity has a finite value (higher than the noise (background) level), completely different discussion has to be given.

4. Optical activity change at the structural transition with substantial symmetry change is rather natural. The authors' explanation is not a fault, but I do not think that this observation is significantly unique.

In conclusion, though the data of optical activity and dielectric properties seem to be acceptable, I think that the total material characterization is insufficient and unacceptable. In addition, the main point of the subject of this manuscript may not have a sufficient scientific impact. Therefore, I do not recommend publication of this manuscript in Nature Communications.

Reviewer #2 (Remarks to the Author):

The authors in this work report a very interesting homochiral organic ferroelectrics with anomalously rotary polarization. The compound exhibits both ferroelectric properties and notable optical rotation during the paraelectric-ferroelectric phase transitions. Thus, present work will be great helpful to rationally design new ferroelectric with interesting optical activities. I'd like to recommend publication of the work in Nature Communications after the following minor revisions.

(1) circular dichroism spectroscopy (CD) is suggested to be performed to the compound even the optical rotation spectrum has been characterized.

(2) Piezoelectric effects is suggested to be characterized for the titled compound.

(3) Chem. Mater., 2015, 27(12), 4493; Angew. Chem., Int. Ed., 2016, 55(22), 6545 and Angew. Chem., Int. Ed., 2016, DOI: 10.1002/anie.201606079 are suggested to be cited.

REVIEWERS' COMMENTS:

Reviewer #1 (Remarks to the Author):

The manuscript was considerably improved. As I requested, the authors reanalyzed the X-ray diffraction data, and obtained the results that exactly I suggested (highly symmetric plastic phase for the high temperature phase).

Now the manuscript is readable, and may be subjected to the reviewing. I think that the following points are still to be issued for the further revisions (sentences in the submitted manuscript is in italic).

The authors write the manuscript as if they are the first who noticed the plastic phase transition in the quinuclidinium derivatives that shows ferroelectric–paraelectric phase transition. However, very recently, there is an important precedent. The work is actually cited as reference 27 (J. Harada, et al. Nature Chemistry, 8, 946-952 (2016)), but the authors should mention that their work is the first report that shows the change of molecular reorientational motion in the crystalline phase induces ferroelectric–paraelectric transition. For example, the sentences in p.6,

"The HT single crystal X-ray diffraction shows few peaks with relatively weak intensity especially for those in the relatively high-angle region. These diffraction characteristics remind us the plastic phase, which is characterized by high symmetry (often cubic), and by average structure Bragg reflections, both few in number and weak in intensity, accompanied by a large amount of diffuse scattering due to the severe orientational and/or displacive disorder. The high-temperature structure was....."

should be re-written such as,

"The high temperature single crystal X-ray diffraction shows few peaks with relatively weak intensity especially for those in the relatively high-angle region. We noticed that this situation is very similar to that recently observed for the quinuclidinium salt that undergoes ferroelectric transition [Ref. 27]. In this case, the paraelectric phase (high temperature phase) becomes a cubic plastic phase. At the transition point, isotropic cubic phase changes to a polar rhombohedral phase. In our case, the high-temperature structure was....."

The authors used "HTP" and "LTP" without any definition.

If the high temperature phase is a cubic plastic crystal, one can align the polarization axes by applying an electric field, as demonstrated in ref. 27. The authors have tried such poling experiments? This could give a larger remnant polarization.

In p.7, the authors should add a reference for the sentence of *"at the level for those of molecular plastic conductors"*.

Since the low temperature phase is SHG active, I do not think that the most of the sentences in p.8 are not necessary. Please remove the following parts.

"SHG is described by the third rank polar property tensor $\chi(2)$, analogous to the piezoelectric coefficient tensor, which vanishes in the 11 centrosymmetric point groups and the noncentrosymmetric point group 432. This is due to the restriction by point group symmetry, since all components of the $\chi(2)$ tensor are zero in these SHG-inactive point groups. For the 20 piezoelectric point groups, however, only 18 of them can allow the appearance of SHG response under the restriction of Kleinman's symmetry. The matrices of other two point groups 422 and 622 are zero. The two point groups require two nonvanishing components of $\chi(2)$ to follow the equation = -. Furthermore, to satisfy the requirement of all symmetry transformations, must be equal to . These

lead to and equal to zero. This means that only the other 18 point groups are SHG-active. To derive symmetry information of the HTP, we measured the variable-temperature SHG response.⁴⁵ For point group 4, there are four nonvanishing second-order susceptibility tensors (χ_{111} , χ_{112} , and χ_{113}) under the restriction of Kleinman's symmetry,³³ only two are independent (χ_{111} , χ_{112}). The matrix is given as

Please move the following sentences in p. 19 to the section of "methods".

The first-principles calculations were performed within the framework of density functional theory (DFT) implemented in the Vienna ab initio Simulation Package (VASP).^{45, 46} The energy cut-off for the expansion of the wave functions was fixed to 550 eV and the exchange–correlation interactions were treated within the generalized gradient approximation of the Perdew–Burke–Ernzerhof type.⁴⁷ For the integrations over the k-space we used a 4x4x1 k-point mesh. The experimental room temperature crystal structure was used as the ground state for evaluating the ferroelectric polarization. In order to evaluate the ferroelectric polarization, we consider the path connecting the non-polar to the polar structure by linearly interpolating the atomic positions.

The manuscript may be acceptable for publication, if the above points are properly revised.

Reviewer #2 (Remarks to the Author):

The raised points have been well solved. The work is very important and the results are convincing. Thus, I'd like to recommend publication of the work in Nature Communications.

Response

(Our responses are marked in blue, those from the main text in purple)

Reviewers' comments:

Reviewer #1 (Remarks to the Author):

This manuscript describes the optical rotation switching in homochiral organic ferroelectrics. The main point of this manuscript may be the finding of the optical activity change at the ferroelectric transition. The authors presented the structural, optical, and dielectric data, but I think that there are several serious faults and insufficiency in their interpretations.

1. The powder X-ray diffraction of HTP shows sharp peaks even in the relatively high-angle region. Smaller number of peaks observed means that the symmetry of the unit cell becomes very high. What I could not really understand is why the authors did not attempt to index the powder pattern. For such a simple pattern, it is rather easy to index the peaks. I think that the authors should index the pattern and determine the lattice parameters (crystal system, cell volume and the Z value) based on the powder diffraction of HTP. This information is extremely important for further discussion.

Reply: We really appreciate the reviewer for question 1 and 2 which reminds us the plastic phase of the high temperature phase (HTP). In the first version, we just focus on whether the HTP is a paraelectric phase or not, and thus ignore more structural information which can be deduced from the X-ray diffraction. In this revision, we not only indexed the PXRD, but also collected the high-quality high-temperature single-crystal diffraction data and thus determined the high-temperature structure. The results from the powder and single crystal are consistent. By combination the basic ferroelectric rule (Aizu rule), we find the determined space group is just the only permissible one (see the following text marked in purple from the main text). With the high temperature structure, all the properties and details can be well explained, including the ferroelectric mechanism, including the symmetry breaking, the SHG and piezoelectric effect transition, optical rotation.

The added the related results and discussion are as following:

To probe the high-temperature structural information, both the variable-temperature single crystal and powder X-ray diffractions were measured (Figure S3, Supplementary Information). Patterns of the PXRD (powder X-ray diffractions) below T_c match well with the simulated one from the single crystal structure, revealing the high crystallinity and purity of the phase (Figure S4, Supplementary Information). Many diffractions peaks observed at below T_c disappear upon heating above T_c . The smaller number of peaks observed means that the symmetry of the HTP becomes very high. The

PXRD data at 363 K were refined with the Pawley method (Figure S5, Supplementary Information), a cubic unit cell with $a = 9.4666 \text{ \AA}$ was suggested, and the most possible space group is the $F432$.

The HT single crystal X-ray diffraction shows few peaks with relatively weak intensity especially for those in the relatively high-angle region. These diffraction characteristics remind us the plastic phase, which is characterized by high symmetry (often cubic), and by average structure Bragg reflections, both few in number and weak in intensity, accompanied by a large amount of diffuse scattering due to the severe orientational and/or displacive disorder.^{31, 32} The high-temperature structure was refined in the space group $F432$, $a = 9.5084(18) \text{ \AA}$. The Pattern of the PXRD matches well with the simulated one from the single crystal structure (Figure S5, Supplementary Information). The relationship of the two temperature cells is $\mathbf{a}^{\text{HTP}} \approx \mathbf{a}^{\text{LTP}} + \mathbf{b}^{\text{LTP}}$, $\mathbf{b}^{\text{HTP}} \approx -\mathbf{a}^{\text{LTP}} + \mathbf{b}^{\text{LTP}}$, $\mathbf{c}^{\text{HTP}} \approx 0.5\mathbf{c}^{\text{LTP}}$ (Figure 1 a). The 4_2 fourfold screw axis of the HTP along the c -direction become the 4_1 fourfold screw axis of the LTP because of the doubling of the c -axis upon the transition from the HTP to LTP, and accordingly, the origin of the LTP cell is at the position (0.25, 0.25 0) of the HTP cell. The crystal packing of the HTP is similar to that of the LTP (Figure 1b). The cation is located at the crystallographic special position 432, and thus is severely disordered and molded with a spherical structure. It can be considered to undergo a high degree of dynamic reorientation (rotation). This suggests that the HTP is a plastic phase in which the molecules are nearly freely rotating. We recorded the conductivity of **1** in the temperature range from 300 to 373 K (Figure S6, Supplementary Information). The conductivity indeed changes from room-temperature 4.27×10^{-6} to high-temperature $1.99 \times 10^{-4} \text{ S/m}$ (at the level for those of molecular plastic conductors) at around T_c , verifying the plastic nature of the HTP.

Figure 1 | The comparison of the crystal structures of the LTP and HTP, showing the similarities in packing and the differences in orientation states of the cations. (a) A view of a [1 1 0] layer of the LTP. The cell edges of both the LTP and HTP were drawn to show their relationship. (b) A view of a [0 1 0] layer of the HTP.

In general, it's not easy to determine the space group of a plastic phase reliably because the above-mentioned X-ray diffraction characteristics. In this case, the space group determined from the X-ray diffraction is just the one and the only one satisfying symmetry requirement for the ferroelectric phase transition. The ferroelectric phase has the point group 4. According to Aizu rule (see Table I in Ref.⁴²), there are only two possible paraelectric point groups: 422 and 432. Since the both powder and single crystal X-ray diffractions clearly reveal FCC lattices of the HTP, the paraelectric point groups should be 432. In addition, the observation that the optical axis sometimes changes upon the transitions from ferroelectric→paraelectric→ferroelectric phase excludes the point group 422 as the paraelectric point group, because the 422F4 ferroelectric species is uniaxial, and the polar axis, which is just the optical axes, cannot change upon the transitions. Contrary to the 422F4

ferroelectric species, the 432F4 ferroelectric species is multiaxial, which allow the change of the polar axis (optical axis) upon the phase transitions. There are just two space groups which belong to the FCC lattice and point group 432: $F432$ and $F4_132$. It's worth noting that the c -axis of the paraelectric phase is the doubling of that of ferroelectric phase. This require that paraelectric phase should have the 4_2 fourfold screw axis, not the 4_1 fourfold screw axis. Only the space group $F432$ contains the 4_2 fourfold screw axes. The 4_2 fourfold screw axis of the space group $F432$ becomes the 4_1 fourfold screw axis of the space group 4_1 upon the transition from the paraelectric to ferroelectric phase, which determines not only the \mathbf{c}^{LTP} , but also the \mathbf{a}^{LTP} and \mathbf{b}^{LTP} as well as the origin which is located on the 4_1 fourfold screw axis. That's why the relationship of the two temperature cells is $\mathbf{a}^{\text{HTP}} \approx \mathbf{a}^{\text{LTP}} + \mathbf{b}^{\text{LTP}}$, $\mathbf{b}^{\text{HTP}} \approx -\mathbf{a}^{\text{LTP}} + \mathbf{b}^{\text{LTP}}$, $\mathbf{c}^{\text{HTP}} \approx 0.5\mathbf{c}^{\text{LTP}}$ (Figure 1 a). Therefore, the crystal of **1** is of the 432F4 ferroelectric species among the 88 ferroelectric species.

2. If HTP has a high-symmetry lattice and contains “severe disorder”, the molecules are considered to undergo a high degree of dynamic reorientation (rotation). This suggests that the HTP is a plastic phase in which the molecules are nearly freely rotating. For such a case, electron density becomes featureless (spherically distributed electrons at the lattice sites), and the molecular framework and chirality may be vanished. Enantiomorphic point groups may be applicable only when the chirality in the lattice is warranted (This is the case when the site symmetry derived from the point 1 is consistent with the molecular chirality. In this case, electron density can be reproduced by superimposing multiple orientations of the molecular framework (restricted reorientation)). Because the authors should discuss the structure of HTP taking the results obtained from the point 1 into account, the present point group assignment for HTP being persistent in non-centrosymmetric may be fundamentally wrong.

Reply: We agree the reviewer with the viewpoint on the relationship between crystal symmetry and the molecular chirality. Therefore, we avoid to deduce the high-temperature symmetry from the molecular chirality in the manuscript, but from the X-ray diffraction as suggested by the reviewer. Accordingly, we delete Table 1 and related discussion. As replied in question 1, the high-temperature space group can be exclusively determined from the X-ray diffraction.

3. Though the authors mention “the SHG signal undergoes a clear transition from non-zero to zero intensity” in p. 7, the SHG intensity in HTP shown in Figure 2b is not zero. This made the reviewer in confusion. If the SHG intensity has a finite value (higher than the noise (background))

level), completely different discussion has to be given.

Reply: In this revision, we calibrated our instrument, and re-measured the variable-temperature SHG. The result is shown in Figure R1. Since our structure analysis definitely reveals the 432F4 phase transition, we think the result is reasonable. The result is included in the revised manuscript.

Figure R1. The result of variable-temperature SHG.

4. Optical activity change at the structural transition with substantial symmetry change is rather natural. The authors' explanation is not a fault, but I do not think that this observation is significantly unique.

Reply: We agree that the symmetry requirements for optical activity transitions have been well known, and the phenomena have been observed in many cases including ferroelectric crystals. However, as far as we know, the past investigations on optical activity change were focused mainly on the optical activity crystals without chiral molecules. Currently, the design of molecular ferroelectrics to realize novel functions have attracted great interest. Optical activity of the recent emerging molecular ferroelectrics hasn't been reported before, but it is really important for understanding the ferroelectricity. For example, we observed in this case that the optical axis can be changed because the ferroelectric crystal is multiaxial. The integration of optical activity into molecular ferroelectrics would add the degree of freedom for device design. In addition, as reminded by the reviewer that both the ferroelectricity and the optical activity change are related to the plastic phase transition, we are aware that it is interesting that in the optically active point group (432), the optically active molecules act as in the solution.

In conclusion, though the data of optical activity and dielectric properties seem to be acceptable, I think that the total material characterization is insufficient and unacceptable. In addition, the main point of the subject of this manuscript may not have a sufficient scientific impact. Therefore, I do not recommend publication of this manuscript in Nature Communications.

Reply: As answered in question 1, the high-temperature structure was determined, all the observed phenomena/properties can be well accounted for. All ferroelectric properties are well done and as we are aware, this is very unique homochiral ferroelectric example never found before. Extraordinarily this phase transition can be attributed to plastic phase transition with the help of the excellent reviewer and is unprecedented. So we do believe that this work can be published in the esteemed journal Nature Communication.

Reviewer #2 (Remarks to the Author):

The authors in this work report a very interesting homochiral organic ferroelectrics with anomalously rotary polarization. The compound exhibits both ferroelectric properties and notable optical rotation during the paraelectric-ferroelectric phase transitions. Thus, present work will be great helpful to rationally design new ferroelectric with interesting optical activities. I'd like to recommend publication of the work in Nature Communications after the following minor revisions.

(1) circular dichroism spectroscopy (CD) is suggested to be performed to the compound even the optical rotation spectrum has been characterized.

Reply: As suggested, we first measure the Uv-vis in the wavelength range of 200 to 800 nm. No absorption peak was observed. This is reasonable since the molecule contains saturate single bonds (sp^3 , no conjugation system or double bond). Therefore, CD spectrum will show no absorption peaks. Therefore we can not measure the CD spectrum in commercial CD instrument.

(2) Piezoelectric effects is suggested to be characteized for the titled compound.

Reply: As suggested, we measured the piezoelectric effects. The results and discussion are added as following:

The piezoelectric effect is the similar physical property which depend on the crystal symmetry. We thus also measured piezoelectric effect in the temperature range from room temperature to 373 K. As expected, the d_{33} undergoes a transition from the non-zero to zero at around T_c (Figure S4, Supplementary Information). The results are also consistent with the symmetry requirements of the HTP and LTP.

Figure S4. Temperature-dependent piezoelectric coefficient d_{33} for **1**.

(3) Chem. Mater., 2015, 27(12), 4493; Angew. Chem., Int. Ed., 2016, 55(22),6545 and Angew. Chem., Int. Ed., 2016,DOI: 10.1002/anie.201606079 are suggested to be cited.

Reply: These literatures documented recent findings in molecular ferroelectrics, we cited them properly as suggested.

Point-by-point response

(Our response are marked in blue, those from the main text in purple)

REVIEWERS' COMMENTS:

Reviewer #1 (Remarks to the Author):

The manuscript was considerably improved. As I requested, the authors reanalyzed the X-ray diffraction data, and obtained the results that exactly I suggested (highly symmetric plastic phase for the high temperature phase).

Now the manuscript is readable, and may be subjected to the reviewing. I think that the following points are still to be issued for the further revisions (sentences in the submitted manuscript is in italic).

Reply: We thank the reviewer for the positive assessment of our revision. We also appreciate the reviewer for the valuable suggestions on the technological writing. These suggestions, ranging from the data analysis to the technological writing, can help us improve the quality of the paper indeed. We think we benefit greatly by these suggestions.

The authors write the manuscript as if they are the first who noticed the plastic phase transition in the quinuclidinium derivatives that shows ferroelectric – paraelectric phase transition. However, very recently, there is an important precedent. The work is actually cited as reference 27 (J. Harada, et al. Nature

Chemistry, 8, 946-952 (2016)), but the authors should mention that their work is the first report that shows the change of molecular reorientational motion in the crystalline phase induces ferroelectric–paraelectric transition. For example, the sentences in p.6,

"The HT single crystal X-ray diffraction shows few peaks with relatively weak intensity especially for those in the relatively high-angle region. These diffraction characteristics remind us the plastic phase, which is characterized by high symmetry (often cubic), and by average structure Bragg reflections, both few in number and weak in intensity, accompanied by a large amount of diffuse scattering due to the severe orientational and/or displacive disorder. The high-temperature structure was....."

should be re-written such as,

"The high temperature single crystal X-ray diffraction shows few peaks with relatively weak intensity especially for those in the relatively high-angle region. We noticed that this situation is very similar to that recently observed for the quinuclidinium salt that undergoes ferroelectric transition [Ref. 27]. In this case, the paraelectric phase (high temperature phase) becomes a cubic plastic phase. At the transition point, isotropic cubic phase changes to a polar rhombohedral phase. In our case, the high-temperature structure was....."

Reply: We revised this part as suggested.

The authors used "HTP" and "LTP" without any definition.

Reply: In revised manuscript, we define ferroelectric phase and paraelectric phase as FP and PP as follows, respectively. "HTP" and "LTP" are replaced.

The structural phase transition is one of the most important properties for understanding the ferroelectricity, we first determined the crystal structures of the ferroelectric and paraelectric phases (abbreviated as FP and PP respectively) (for crystallographic information, see Supplementary Data 1–4 and Supplementary Table 1).

As will be described below, the phase below T_c is the FP, the phase above T_c is the PP.

If the high temperature phase is a cubic plastic crystal, one can align the polarization axes by applying an electric field, as demonstrated in ref. 27. The authors have tried such poling experiments? This could give a larger remnant polarization.

Reply: We agree that for multiaxial ferroelectrics, the polarization axes can be aligned by applying an electric field. In ref. 28 of the revised manuscript, the alignment of the polar axis was examined for the powder-pressed sample by measuring the P - E hysteresis loops. In our case, the relatively large coercive field prevent us using the same method to examine the alignment of the polar axis. As described in our main text, the measurement of the P - E hysteresis loop was carried out on a thin film capacitor with the configuration of Au/Sample film/ITO. The alignment of the polar axis for multiaxial ferroelectrics is essentially the same as the polarization reversal, and it can be realized not only by using field cooling but also by applying a field directly. That's why ceramic ferroelectrics with randomly oriented grains can be used as polar materials after poling. Accordingly, we provide

sufficient information on the polarization reversal by using the Sawyer-Tower circuit and PFM. Of course, further investigation on the alignment of the polar axis as described in Ref. 27 would be of interest. We might carry our such work in the future.

In p.7, the authors should add a reference for the sentence of "*at the level for those of molecular plastic conductors*".

Reply: A reference is added. In that literature, the conductivity of the crystalline and plastic phases is compared, the conductivity of $2.6 \times 10^{-4} \text{ S cm}^{-1}$ is reported for the plastic phase.

Since the low temperature phase is SHG active, I do not think that the most of the sentences in p.8 are not necessary. Please remove the following parts.

"SHG is described by the third rank polar property tensor $\chi(2)$, analogous to the piezoelectric coefficient tensor, which vanishes in the 11 centrosymmetric point groups and the noncentrosymmetric point group 432. This is due to the restriction by point group symmetry, since all components of the $\chi(2)$ tensor are zero in these SHG-inactive point groups. For the 20 piezoelectric point groups, however, only 18 of them can allow the appearance of SHG response under the restriction of Kleinman's symmetry. The matrices of other two point groups 422 and 622 are zero. The two point groups require two nonvanishing components of $\chi(2)$ to follow the equation $\chi_{ijk} = -\chi_{ikj}$. Furthermore, to satisfy the requirement of all symmetry transformations, must be equal to χ_{jik} . These lead to $\chi_{111} = \chi_{222} = \chi_{333} = 0$ and equal to zero. This means that only the other 18 point groups are SHG-active. To derive symmetry

information of the HTP, we measured the variable-temperature SHG response.", and
"For point group 4, there are four nonvanishing second-order susceptibility tensors
(, , and) under the restriction of Kleinman' s symmetry,³³ only two are independent
(,). The matrix is given as
."(whole sentence with the matrix)

Reply: As suggested, we remove those.

Please move the following sentences in p. 19 to the section of "methods".

"The first-principles calculations were performed within the framework of density functional theory (DFT) implemented in the Vienna ab initio Simulation Package (VASP).^{45, 46} The energy cut-off for the expansion of the wave functions was fixed to 550 eV and the exchange–correlation interactions were treated within the generalized gradient approximation of the Perdew–Burke– Ernzerhof type.⁴⁷ For the integrations over the k-space we used a 4x4x1 k-point mesh. The experimental room temperature crystal structure was used as the ground state for evaluating the ferroelectric polarization. In order to evaluate the ferroelectric polarization, we consider the path connecting the non-polar to the polar structure by linearly interpolating the atomic positions."

Reply: As suggested, we rearrange this part.

The manuscript may be acceptable for publication, if the above points are properly revised.

Reviewer #2 (Remarks to the Author):

The raised points have been well solved. The work is very important and the results are convincing. Thus, I'd like to recommend publication of the work in Nature Communications.

Reply: We thank the reviewer for the positive assessment of our revision.